# IO-LVM: Inverse Optimization Latent Variable Models with Graph-based Planning Applications

## Abstract

Learning representations for solutions of constrained optimization problems (COPs) with unknown cost functions is challenging, as models like (Variational) Autoencoders struggle to capture constraints to decode structured outputs. We propose an inverse optimization latent variable model (IO-LVM) that constructs a latent space of COP costs based on observed solutions, enabling the inference of feasible and meaningful solutions by reconstructing them with a COP solver in the loop. To achieve this, we leverage estimated gradients of a Fenchel-Young loss through a non-differentiable deterministic solver while shaping the embedding space. In contrast to established Inverse Optimization or Inverse Reinforcement Learning methods, which typically identify a single or context-conditioned cost function, we exploit the learned representation to capture underlying COP cost structures and identify solutions likely originating from different agents or conditions, each using distinct cost functions when making decisions. Using both synthetic and actual ship routing data, we validate our approach through experiments on paths and cycles inference problems, demonstrating the interpretability of the latent space and its effectiveness in path/cycle reconstruction and their distribution prediction.

## 1. Introduction

When learning latent generative representations, it is often necessary for inferred samples to satisfy specific constraints, such as forming paths in a graph between designated start and target nodes. This restriction introduces the challenge of ensuring that the model generates feasible solutions with

respect to a constrained optimization problem (COPs) formulation. The difficulty intensifies when the feasible set of solutions is discrete, as the gradients of these solutions with respect to the model parameters are zero almost everywhere and therefore non-informative (Abbas & Swoboda, 2021).

State-of-the-art approaches to recovering underlying cost functions of COPs from observed solutions, e.g., structured decisions performed by agents, primarily address the non-informative gradient problem by either smoothing solver operations (Lahoud et al., 2024), interpolating COP solutions (Pogančić et al., 2020b), or perturbing the COP cost (Berthet et al., 2020). The alternative are approaches based on a relaxation of the COP, such as Maximum Entropy Inverse Reinforcement Learning (Ziebart et al., 2008b), that seek to match statistics of the observes behavior (i.e., solutions). However, these methods are unable to directly learn from data of multiple different agents with different underlying cost functions because they assume a single underlying cost. To correctly recover several agents from real world data containing behavior from different agents, they require supervision through agent labels.

In this paper, we introduce IO-LVM, a novel approach for learning latent representations of COP costs that can recover observed COP solutions, specifically for path and cycle problems in graphs. Our approach does not assume a single underlying COP cost, allowing it to learn effectively even when multiple agents or context are represented in the data without labels. Similar to a Variational Autoencoder (VAE) (Kingma, 2013), we use amortized inference and map into a meaningful and interpretable low-dimensional latent cost space. In contrast of ordinary VAEs, we guarantee that samples fulfill requirements of the feasible set (e.g., connected paths) by using a black-box COP solver in the generative step. To address the gradient challenge, we adopt a technique similar to that of Berthet et al. (2020), perturbing the input of the black-box solver and employing the Fenchel-Young loss (Blondel et al., 2020) to estimate the gradients of the COP solutions.

IO-LVM not only capture the dataset's path distributions and predicts paths for new start and target nodes, but also addresses the interpretability challenge by learning a low-dimensional latent space for the cost. In this space, similar

---

[1]Anonymous Institution, Anonymous City, Anonymous Region, Anonymous Country. Correspondence to: Anonymous Author <anon.email@domain.com>.

Preliminary work. Under review by the International Conference on Machine Learning (ICML). Do not distribute.

costs are positioned close to each other, offering a more intuitive and interpretable representation of the path-planning process. This low-dimensional latent space enables new possibilities for behavior cost analysis, such as clustering similar COP costs, denoising observed paths by finding a small number of representative paths, and reconstructing structured solutions. Additionally, IO-LVM allows for predicting how different agents might navigate between unseen source and target nodes, providing a flexible framework for path inference.

### 1.1. Our Contributions

- We introduce IO-LVM, a method that combines variational approximation techniques with COP solver gradient estimation to learn latent representations for the underlying costs of COPs based on observed decisions.

- IO-LVM naturally constructs a disentangled, and sometimes multimodal, latent space, allowing for the reconstruction of observed path distributions without making assumptions about inferred paths. Notably, the ability to recover distinct (e.g., multimodal) representations for the underlying costs enables the modeling of different agents making decisions.

- We demonstrate the versatility of IO-LVM using both synthetic and real-world ship path datasets, highlighting its potential for path analysis tasks such as naturally clustering paths and cycles into meaningful groups, denoising observed paths, and predicting paths for unseen start and target nodes. Our aim is not only to provide quantitative results but also to offer insights through visualizations of paths and latent variables

### 1.2. Related Work

To address the aforementioned gradient challenge, several works have focused on differentiating through convex solvers (Amos & Kolter, 2017; Agrawal et al., 2019), enabling the construction of end-to-end learning frameworks that learn from decisions formulated as solutions to linear or quadratic programs (Donti et al., 2017; Wilder et al., 2019). However, these methods are mainly limited to continuous COP formulations and are difficult to extend to combinatorial problems such as the graph-based problems in our work.

In addition to convex solvers, efforts to differentiate through dynamic programming algorithms have also been explored. For example, Mensch & Blondel (2018) proposed a method that addresses dynamic programming differentiability. Specifically for path inference, Lahoud et al. (2024) proposed a differentiable version of the Floyd-Warshall algorithm to learn from observed paths in graphs. Different to our work, their approach struggles with scalability as graph size increases.

Inverse Optimization (Aswani et al., 2018; Tan et al., 2019; 2020), Inverse Reinforcement Learning, and Inverse Path Planning (Wulfmeier et al., 2017; Lahoud et al., 2024) also learn representations from the solutions of COPs in form of cost parameters and assume that the observed solutions were generated by some optimization process. Their cost parameters are either global (Lahoud et al., 2024), linear (Ng et al., 2000; Ziebart et al., 2008a;b; Nguyen et al., 2015) or nonlinear (Finn et al., 2016; Wulfmeier et al., 2017; Fernando et al., 2020), often learned with end-to-end gradient estimation, exploiting insight in the decision making process. Other methods instead assume a black box optimizer and estimate gradients with respect to its inputs (Pogančić et al., 2020a; Berthet et al., 2020). Similar to our approach, one of the ideas of Berthet et al. (2020) is to utilize a Fenchel-Young loss to match inferred and observed paths within a smooth and convex space. Different to us, these methods typically assume a single underlying cost function or condition the cost on a given context, which may not capture the diversity of agent behaviors present in real-world scenarios.

Although autoencoders (Hinton & Salakhutdinov, 2006) and Variational Autoencoders (VAEs) (Kingma, 2013) have been successful in learning latent representation to facilitate feature extraction, they typically struggle to decode structured outputs, which is essential for path inference tasks. A work with similar motivation to ours is that of Bentley et al. (2022), which combines VAEs with genetic algorithms. However, their method lacks a guarantee of optimality for COPs. In contrast, IO-LVM leverages gradient estimation through a specialized solver, ensuring optimality and feasibility, resulting in a more robust end-to-end learning framework.

## 2. Preliminaries

Below, we recap the Evidence Lower Bound (ELBO) for deep latent variable models and introduce Fenchel-Young losses which are both fundamental for our approach.

### 2.1. Evidence Lower Bound (ELBO)

The objective in latent variable models is to identify the latent variables $\mathbf{z}$ that best explain the observed data $\mathbf{x}$. However, directly computing the posterior $P(\mathbf{z} \mid \mathbf{x})$ is generally intractable. To address this, a variational distribution $q_\phi(\mathbf{z} \mid \mathbf{x})$ is introduced and learned with a lower bound objective (ELBO) (Kingma, 2013; Rezende et al., 2014). The ELBO makes a trade-off between accurately reconstructing the input data (the expected log-likelihood) using a model $p_\theta(\mathbf{x} \mid \mathbf{z})$ and adhering to the prior distribution $P(\mathbf{z})$ for the

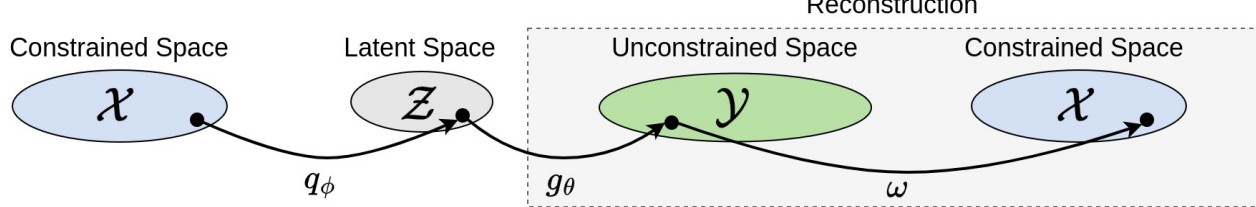

*Figure 1.* Proposed latent space model with constrained reconstruction: structured data is mapped from $\mathcal{X}$ to latent space $\mathcal{Z}$, then reconstructed in two steps—$\mathcal{Z}$ to unconstrained space $\mathcal{Y}$, and $\mathcal{Y}$ to constrained space $\mathcal{X}$ by a solver $\omega$.

latent variables, i.e.,

$$
\begin{aligned}
l(\theta, \phi) = &- \mathbb{E}_{q_\phi(\mathbf{z}|\mathbf{x})} \left[ \log p_\theta(\mathbf{x} \mid \mathbf{z}) \right] \\
&+ \beta D_{\mathrm{KL}} \left( q_\phi(\mathbf{z} \mid \mathbf{x}) \, \| \, P(\mathbf{z}) \right),
\end{aligned} \tag{1}
$$

where $D_{\mathrm{KL}}$ is Kullback-Leibler (KL) divergence and $\beta$ is a balance factor (Higgins et al., 2017; Burgess et al., 2018),. In our approach detailed in Sec. 3, we also use a learned $q_\phi$ (encoder), but replace the usual reconstruction loss with a Fenchel-Young loss (see below).

## 2.2. Fenchel-Young Losses

Fenchel-Young losses are a class of loss functions that generalize many commonly used losses in machine learning and structured prediction (Blondel et al., 2020; Bao & Sugiyama, 2021) and are derived from the Fenchel conjugate in convex analysis (Boyd & Vandenberghe, 2004). Given an input $\mathbf{x}$, a score vector $\mathbf{y}$, a scoring function $f(\mathbf{y}, \mathbf{x})$, and an optimization problem formulated as $\omega(\mathbf{y}) \in \arg\min_{\mathbf{x} \in \mathcal{C}} \langle \mathbf{y}, \mathbf{x} \rangle$, the Fenchel-Young loss is defined as $l_{\mathrm{FY}}(\mathbf{y}, \mathbf{x}) = f(\mathbf{y}, \mathbf{x}) - f(\mathbf{y}, r(\omega, \mathbf{y}))$, where $r$ is a regularization function. The loss compares the score to that of the regularized output $r(\omega, \mathbf{y})$, encouraging solutions $\omega(\mathbf{y})$ that align to the input $\mathbf{x}$.

We use a variant of the Fenchel-Young loss where the optimization $\omega(\mathbf{y})$ is turned into a stochastic process by adding noise (perturbation) $\epsilon$ to the input. This introduces randomness, smoothing the objective function landscape. The perturbed Fenchel-Young loss is expressed as

$$
l_{\mathrm{FY}}^\epsilon(\mathbf{y}, \mathbf{x}) = f(\mathbf{y}, \mathbf{x}) - f(\mathbf{y}, \hat{\mathbf{x}}_\epsilon), \tag{2}
$$

where $\hat{\mathbf{x}}_\epsilon := r(\omega, \mathbf{y}) = \mathbb{E}_\epsilon \left[ \omega(\mathbf{y} + \epsilon) \right]$, and $\epsilon$ is typically drawn from a distribution such as Gaussian. For a more detailed discussion, refer to Blondel et al. (2020). In our approach, $\mathbf{y}$ is a function of latent variables $\mathbf{z}$ and the estimated gradient $\nabla_{\mathbf{y}} l_{\mathrm{FY}}^\epsilon$ is used to differentiate through a black-box COP solver.

## 3. Method

In this section, we introduce the notations and problem definition (Sec. 3.1), present IO-LVM in a general COP setting

(Sec. 3.2), and discuss specifications and assumptions for path and cycle applications (Sec. 3.3).

### 3.1. Notation and Problem Definition

Our dataset $\mathcal{D} = \{(\mathbf{x}_i, \mathbf{p}_i)\}_{i=1}^N$ consists of structured decision vectors $\mathbf{x}_i \in \mathcal{X}$ in a constrained space $\mathcal{X}$, e.g., connected paths performed by agents in a graph, and corresponding problem requirements $\mathbf{p}_i \in \mathcal{P}$, e.g., start and target nodes for the path. We denote by $\omega$ a black-box solver for the COP that takes cost vectors $\mathbf{y}_i$ and problem requirements $\mathbf{p}_i$ to output an optimal COP solution $\hat{\mathbf{x}}_i = \omega(\mathbf{y}_i, \mathbf{p}_i)$.

The main goal is to model a meaningful low-dimensional representation of COP costs $\mathbf{y}_i \in \mathcal{Y}$ that leads to the observed decision vectors from $\mathcal{D}$. Concretely, we aim to estimate the posterior distribution $P(\mathbf{z} \mid \mathbf{x})$, where $\mathbf{z} \in \mathcal{Z} \subseteq \mathbb{R}^k$ is a latent vector in a space of dimension $k$.

### 3.2. IO-LVM Description

Similar to VAEs, we learn a latent representation $\mathcal{Z}$ using a nonlinear mapping $q_\phi$ to map samples $\mathbf{x}_i$ to the latent space $\mathcal{Z}$, and then reconstruct them back to the constrained space $\mathcal{X}$. Different to VAE models, where reconstruction is done by a decoder network, our reconstruction is non-trivial due to the constraints on the COP solution space $\mathcal{X}$. E.g., $\mathcal{X}$ contains valid paths between specific nodes in a graph. To achieve this, we define our reconstruction as a composition of functions $g_\theta \colon \mathcal{Z} \to \mathcal{Y}$ and $\omega \colon \mathcal{Y} \times \mathcal{P} \to \mathcal{X}$, where the former is a nonlinear map parameterized by $\theta$ and the latter is a solver that is potentially non-differentiable. This sequence of transformations is visualized in Fig. 1.

To learn our IO-LVM, we adapt the VAE's ELBO objective by changing the reconstruction loss (first term) of Eq. (1). We introduce the solver $\omega$ and a suitable distance measure $d$ in $\mathcal{X}$ resulting in the term

$$
\mathbb{E}_{q_\phi(\mathbf{z}|\mathbf{x})} \left[ d(\mathbf{x}, \omega(\mathbf{y}^\theta)) \right], \tag{3}
$$

where $\mathbf{y}^\theta := g_\theta(\mathbf{z})$. In contrast to VAE models, the black box solver $\omega$ in our reconstruction generally prohibits end-to-end learning with a common loss such as the Mean

Squared Error. For this reason, we use the Fenchel-Young loss (see Sec. 2.2) for $d$ by inducing perturbations in the input space of the COP. Consequently, our loss function is defined as:

$$l(\theta, \phi) = \mathbb{E}_{q_\phi(\mathbf{z}|\mathbf{x})} \left[ l_{\text{FY}}^\epsilon(\mathbf{y}^\theta, \mathbf{x}) \right] \\ + \beta D_{\text{KL}} \left( q_\phi(\mathbf{z}|\mathbf{x}) \, \| \, P(\mathbf{z}) \right). \quad (4)$$

By choosing $f$ (in $l_{\text{FY}}^\epsilon$) to be a linear cost function, i.e., $f(\mathbf{y}, \mathbf{x}) = \langle \mathbf{y}, \mathbf{x} \rangle$, the gradient of Eq. (2) with respect to $\mathbf{y}$-elements is analytically computed as $\nabla_\mathbf{y} l_{\text{FY}}^\epsilon(\mathbf{y}, \mathbf{x}) = \mathbf{x} - \hat{\mathbf{x}}_\epsilon$, minimizing the Fenchel-Young loss if and only if $\mathbf{x} = \hat{\mathbf{x}}_\epsilon$ (Berthet et al., 2020). With this, the reconstruction loss of Eq. (4) can be rewritten as

$$\mathbb{E}_{q_\phi(\mathbf{z}|\mathbf{x})} \left[ \langle \mathbf{y}^\theta, \mathbf{x} \rangle - \langle \mathbf{y}^\theta, \hat{\mathbf{x}}_\epsilon^\theta \rangle \right], \quad (5)$$

where $\hat{\mathbf{x}}_\epsilon^\theta = \mathbb{E}_\epsilon \left[ \omega(\mathbf{y}^\theta + \epsilon, \mathbf{p}) \right]$. This loss allows us to obtain gradient estimates w.r.t. the weights $\theta$, as the chain of gradients in the reconstruction block can now be written as

$$\nabla_\theta l_{\text{FY}}^\epsilon(\mathbf{y}^\theta, \mathbf{x}) = (\mathbf{x} - \hat{\mathbf{x}}_\epsilon^\theta) \frac{\partial g_\theta(\mathbf{z})}{\partial \theta}. \quad (6)$$

Estimating the gradients in Eq. (6) is generally done in a Monte Carlo fashion, which is expensive due to the need of running the solver $\omega$ several times. Therefore, using the property of expectation linearity, we rewrite the reconstruction loss in Eq. (5) as

$$\mathbb{E}_{q_\phi(\mathbf{z}|\mathbf{x})} \mathbb{E}_\epsilon \left[ \left[ \langle \mathbf{y}^\theta, \mathbf{x} \rangle - \langle \mathbf{y}^\theta, \omega(\mathbf{y}^\theta + \epsilon, \mathbf{p}) \rangle \right] \right], \quad (7)$$

highlighting that the estimator is unbiased with a double expectation. This result allows us to use a Stochastic Gradient Descent (SGD) method to learn the parameters $\theta$ and $\phi$, as described in Alg. 1. By leveraging SGD, the solver runs once (instead of several times) per data sample during training.

**Algorithm details.** The algorithm details the steps in the training process using an encoder $h_\phi$ to model $q_\phi(\mathbf{z} \mid \mathbf{x})$, and a mapping $g_\theta$. Note that in step 1, the problem requirement can be leveraged into the encoder as additional information. Step 2 samples the latent value using the VAE re-parametrization trick (Kingma et al., 2015). In step 3, we include a transformation $\Phi$ to ensure that costs $\mathbf{y}^\theta$ fits to COP input space $\mathcal{Y}$. It is common that some COPs require their cost elements to be positive, for instance, which is generally fixed by ordinary activation functions. In step 4, we compute the COP solution given a inferred and perturbed cost. In step 6, the back-propagation is allowed due to the gradient estimator described in Eq. (6), bridging the gap of the non-differentiable COP solver.

Once the IO-LVM is trained, we can reconstruct paths from parts of the low-dimensional latent space using a composition of $\Phi(g_\theta)$ and $\omega(.)$. Sampling from different parts of

---

**Algorithm 1** One epoch of IO-LVM through a SGD optimization

1: **Components:**
2:    - Encoder $h_\phi$; Decoder $g_\theta$.
3: **Input:** Dataset $\mathcal{D} = \{(\mathbf{x}_i, \mathbf{p}_i)\}_{i=1}^N$
4: **Output:** Trained model parameters
5: **for** each sample $(\mathbf{x}, \mathbf{p}) \in \mathcal{D}$ **do**
6:    **Step 1:** Encoding: $(\mu, \sigma) = h_\phi(\mathbf{x}, \mathbf{p})$.
7:    **Step 2:** Sample $\mathbf{z}$: $\mathbf{z} = \mu + \sigma \cdot \epsilon, \epsilon \sim \mathcal{N}$.
8:    **Step 3:** Map $\mathbf{z}$ to COP cost space: $\mathbf{y}^\theta = \Phi(g_\theta(\mathbf{z}))$.
9:    **Step 4:** Solve the COP using $\omega$, $\mathbf{p}$ and the inferred cost $\mathbf{y}^\theta$: $\hat{\mathbf{x}}_\epsilon^\theta = \omega(\mathbf{y}^\theta + \epsilon, \mathbf{p})$, where $\epsilon \sim \mathcal{N}$.
10:    **Step 5:** Compute the loss as in Eq. (5).
11:    **Step 6:** Update the encoder and decoder parameters $(\phi, \theta)$ allowed by Eq. (6).
12: **end for**

---

the latent space allows us to observe different patterns reconstructed in the path space. In Sec. 4.5, we show that the selection of $\beta$ also mitigates the issue of posterior collapse, which is often encountered in VAE models with powerful decoders (Van Den Oord et al., 2017).

### 3.3. IO-LVM Assumptions and Applications

Since our reconstruction block contains a black-box solver, we assume the observed structured decision samples in $\mathcal{D}$ presented in Sec. 3.1 are COP optimal solutions, e.g., agents perform decisions optimally based on their own underlying cost values. Therefore, the variations in the observed structured decision samples arises from differences in valuations of COP costs. Below, we describe how we model the two problem domains used in experiments in Sec. 4.

**Path Planning.** For a fixed directed graph with edge set $E$, we can model a set of paths as a set of binary vectors $\mathcal{X} \subseteq \{0, 1\}^{|E|}$ corresponding to edge usage. Here, $\mathcal{D}$ contains samples of paths (without cycles) in the graph. In this case, a common path requirement is $\{\mathbf{p} = (s, t) \mid s, t \in V, s \neq t\}$, defining start and target nodes of those paths. In this scenario, we assume there is an underlying set of edges costs for each data sample such that a Shortest Path Problem (SPP) solver (e.g., Dijkstra) $\omega$ recovers the observed paths.

**Hamiltonian Cycles.** For a fixed, either directed or undirected graph, with edge set $E$, we can model a set of cycles also as a set of binary vectors $\mathcal{X} \subseteq \{0, 1\}^{|E|}$ corresponding to edge usage. Here, $\mathcal{D}$ contains samples of Hamiltonian cycles in the graph. Here, we assume that there is an underlying set of edge costs for each data sample such that a Traveling Salesman Problem (TSP) solver recovers the observed cycle. Although path requirements could be added, we consider a null set in our experiments.

Note that in both applications, as in general discrete problems, the COP can be formulated with a linear objective: $\hat{\mathbf{x}} \in \operatorname{argmin}_{\mathbf{x} \in \mathcal{X}} \langle \mathbf{y}, \mathbf{x} \rangle$, fulfilling the requirement for the gradient estimator.

# 4. Experiments

The experiments focus on path planning in graphs using Dijkstra and a TSP solver for the path planning and Hamiltonian cycles assumptions, respectively, as described in Sec. 3.3. Details of the IO-LVM training process is provided in Appendix B. We use three different datasets to evaluate our IO-LVM, including path reconstruction, path distribution prediction, facilitation to latent space analysis, and its potential for unsupervised learning tasks.

## 4.1. Datasets

**Synthetic Waxman Random Graph.** We generate a Waxman graph (Van Mieghem, 2001) with 700 nodes and 7230 edges; with three different cost functions for the edges costs $\mathbf{y}$, all of them on the basis of a nonlinear function from unobserved features. We increased the costs of southern edges for agent 1 and of northern edges for agent 3, while agent 2 was unbiased in terms of south/north edges (e.g., in in the top-left graph in Fig. 2 it is clear that agent 1 prefers traveling through north edges)). We solved the SPP using Dijsktra for each agent cost multiple times on the basis of the generated edges costs plus a Gaussian noise (i.e., $\omega(\mathbf{y} + \epsilon)$) to generate multiple paths to $\mathcal{D}$. These paths are generated in two manners, one set with a single source-target pair (Fig. 2, top-left) and another set with multiple pairs (Fig. 2, bottom-left). In both cases, 6,000 paths were generated, which 5,000 were used for training. More details in Appendix A.

**Ships Dataset.** Using Automatic Identification System (AIS) data from the Danish Maritime Authority (Danish Maritime Authority, 2020), we use ship locations collected during three months. We project the locations to a grid graph with 2513 nodes and 8924 edges, resulting in a set of 2,500 paths with different start and target nodes, where 2,000 were used for training. More details in Appendix A.

**TSPLIB** From TSPLIB95 (Reinelt, 1991), we use *burma14* (14 nodes, 91 edges) and *bayg29* (29 nodes, 406 edges) graphs. Actual (underlying) edges costs $\mathbf{y}$ are generated using a nonlinear function incorporating unobserved features and Euclidean distances between nodes as offset. We create two versions for each graph, one using 3 unobserved features and another using 50 unobserved features. 3,000 actual Hamiltonian Cycles (2,400 for training) are generated with a TSP solver $\omega$ using the underlying costs $\mathbf{y}$ as input. More details in Appendix A.

## 4.2. Latent Space Analysis

In this experiment, we analyze the projection of 1,000 test paths to the latent space $\mathcal{Z}$ using the encoder $h_\phi$. The goal of this experiment is to observe that paths generated by similar costs (e.g., coming from the same agent or similar context) are projected close to each other in latent space after IO-LVM training.

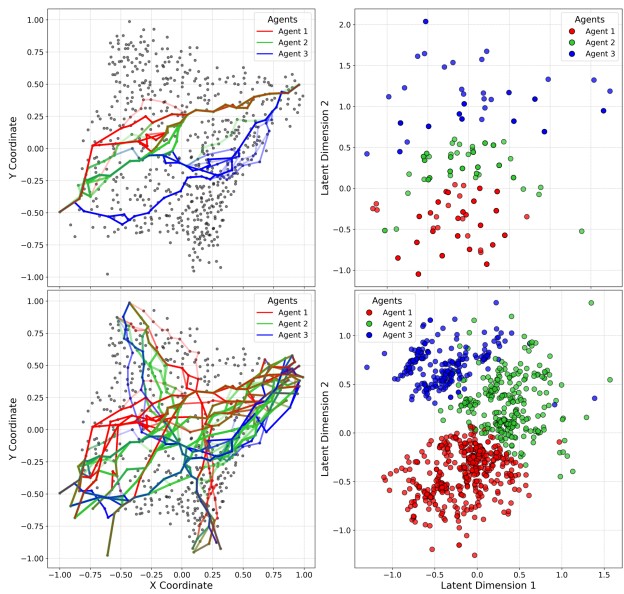

*Figure 2.* Left charts illustrate the paths dataset for single (top) pair of start and target nodes and for multiple (bottom) pairs of start and target nodes. Right charts illustrate the respective latent space embedding of paths after training.

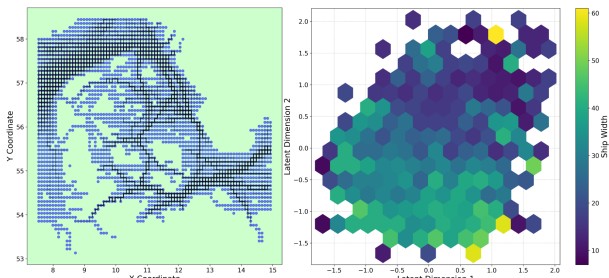

*Figure 3.* Ship paths in the left chart are represented in black, and are projected to the latent space using $q_\phi$. The colors in the right chart represent the average ship width in each hexagon of the latent space. The ship width, although not used in training, are observed as a correlated feature in the latent space.

**Synthetic Waxman Paths.** Fig. 2 illustrates the two dim. latent space. The colors indicate which of the three agents performed the task. Note that the agent identity was not provided in training. We observe that IO-LVM successfully disentangles the factors associated with the costs of the

three different agents. This disentanglement is evident not only when the dataset contains observed paths between a single pair of start and end nodes (Fig. 2, top-right) but more importantly when several different pairs of start and end nodes are present (Fig. 2, bottom-right). The example with multiple pairs is important because it highlights that IO-LVM is capable of encoding the underlying transition costs (and not only paths) if there is enough data. Note that there are, for example, multiple different paths performed by Agent 1 (red), even with different start and target nodes, but almost all these paths are mapped into the same region in the latent space because they share similar underlying transition costs.

**Ship Paths.** The left chart in Fig. 3 illustrates the 2D latent space (3D latent space results are seen in Fig. 7 in the Appendix). Each hexagon in the right chart of Fig. 3 corresponds to a subspace of the latent space. For each hexagon, the average of the ships' width is plotted in color. Larger ships are less frequently found in the top-right corner of the graph, leading to a low average ship width in that region. This is another example that IO-LVM was capable to capture unobserved factors within the latent space, i.e., the ship width information (provided by AIS) was not used during the training process.

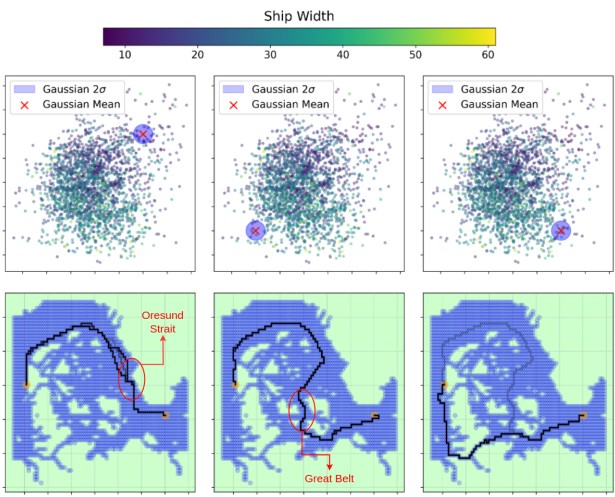

Figure 4. Reconstruction for the ship dataset. Top charts: region of samples from a Gaussian in the latent space. Bottom charts: corresponding generated trajectories in the graph given a hypothetical (non-existent in the training paths) pair of start and target nodes.

**Hamiltonian Cycles.** We select nine samples from the validation set of the *burma14* graph, distributed across different regions of the latent space. They are organized into three groups of three samples each (see the top graph of Fig. 11 in Appendix). The corresponding paths that generated these latent values are visualized in the bottom graph of Fig. 11. Paths with a more edges intersection tend to be

closer to each other in the latent space. We generalize this analysis by computing the Euclidean distance between all pairs of latent values versus the Manhattan distance based on edge usage (path choice) between those paths. For each group of sample pairs with a specific Manhattan distance, we calculate the average Euclidean distance in the latent space. The results, presented in Fig. 10 in Appendix, reveal that samples that are closer in the latent space are also closer in terms of edge intersection.

### 4.3. Qualitative Reconstruction Analysis

In this experiment, we analyze how structure in the latent space influences reconstruction. For this, we sample 20 latent cost functions from Gaussians in latent space and compare distribution of paths generated by the composition of the decoder and solver.

**Synthetic Waxman Paths.** Reconstructed synthetic paths are shown in the bottom graphs of Fig. 12 (Appendix). It is observed that points closer in the latent space share a high number of edges in the graph. Additionally, as the variance increases, the number of distinct reconstructed paths grows, indicating consistency in the learned latent space. E.g., difference between the third and fourth columns in Fig. 12.

**Ship Paths.** As seen in Fig. 4, neighbor latent values share a high number of edges in the graph (e.g., many path samples are the same). Moreover, an interesting pattern emerges: some regions of the latent space containing wider ships avoid the Oresund Strait when traveling from the east to the north part of Denmark even though it is the shortest path in terms of euclidean distance, as observed in the second column of the figure where ships prefer going through the Great Belt. Note that Dijsktra in the reconstruction ensures that all reconstructed paths remain feasible.

**Hamiltonian Cycles.** We input Hamiltonian cycles to the encoder to evaluate whether they are correctly reconstructed. Fig. 5 illustrates some samples reconstruction of IO-LVM against VAE , where the decoder outputs paths as binary edge usage indicators (i.e., probabilities converted to binary). In the figure, thick edges illustrate the reconstructed paths for the first three validation samples of each dataset (*burma14* and *bayg29*). It can be seen from the figure that, different from VAEs, IO-LVM ensures that the output forms a valid Hamiltonian cycle due to the inclusion of a TSP solver in its processing loop. This happens even when the reconstruction is not fully correct (e.g., most right graph in in Fig. 5). Reconstructions of other baselines and the respective ground truth are provided in the Appendix (Fig. 13). There, we include the TSP-EUC as a naive, non-learning baseline where the TSP solution is computed using edge costs defined as the Euclidean distances between node po-

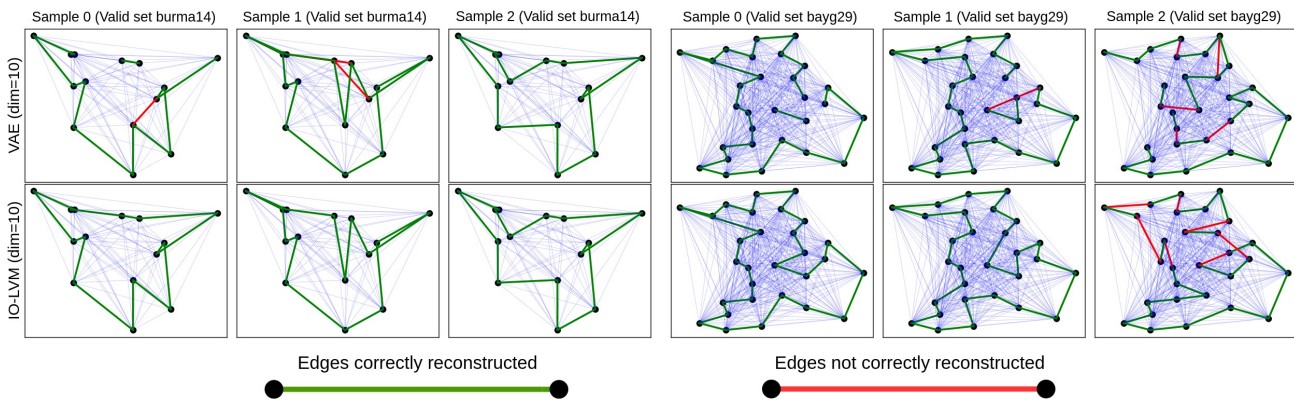

*Figure 5.* Each column illustrates an inferred sample for the Hamiltonian Cycles experiment using VAE (top) and IO-LVM (bottom) with 10 latent dimensions on the first three samples of each path dataset generated with 50 dimensions. Green edges denote correct reconstructions relative to the validation path, while red edges indicate false positives. VAEs might yield unstructured outputs, not being able to guarantee Hamiltonian cycles reconstruction. Groundtruths and other baseline results are available in the Appendix, Figure 13 .

*Table 1.* Reported are the average Recall of edges reconstruction on the train/validation set for the Hamiltonian Cycles experiment.

| Methods | Latent Dims | burma14 (3 dims) | bayg29 (3 dims) | burma14 (50 dims) | bayg29 (50 dims) |
|---------|-------------|------------------|-----------------|-------------------|------------------|
| TSP-EUC | – | 0.634/0.636 | 0.557/0.552 | 0.535/0.532 | 0.596/0.590 |
| VAE | 2 | 0.852/0.835 | 0.660/0.639 | 0.626/0.582 | 0.521/0.500 |
| VAE | 10 | 1.000/0.958 | 0.969/0.891 | 0.999/0.884 | 0.877/0.741 |
| IO-LVM | 1 | 0.839/0.823 | 0.739/0.729 | 0.679/0.663 | 0.636/0.620 |
| IO-LVM | 2 | 0.928/0.892 | 0.854/0.799 | 0.846/0.763 | 0.751/0.653 |
| IO-LVM | 10 | **1.000/0.976** | **0.999/0.939** | **1.000/0.939** | **0.997/0.836** |

sitions; and IO-LVM and VAE with less number of latent dimensions.

### 4.4. Quantitative Results

Here, we aim to measure the reconstruction and prediction power of IO-LVM versus selected baselines. In the reconstruction experiment, the idea is to understand if IO-LVM is capable to reconstruct the Hamiltonian cycles input of both training and validation cycles with a limited number of latent dimensions. In the predictive experiment, we aim to measure the prediction quality of the overall validation paths distribution, where we do not input the observed paths in the encoder, but used a kernel density estimator (KDE) in the learned latent space to estimate the probability density function and sample predictions from it.

**Reconstruction of Hamiltonian Cycles: Metric and Results.** Reconstruction performance is evaluated using Recall of edges usage. The choice of this metric in the Hamiltonian experiment is due to the fixed number of actual edges usage (true positives plus false negatives). Table 1 demonstrates that IO-LVM outperforms VAE for a fixed number of latent dimensions. The superior performance of IO-LVM is attributed to its structured reconstruction process, which

is not guaranteed in VAEs. Note that even though the reconstruction results of VAE with 10 latent dimensions are reasonable, the mistakes are generally non-structured outputs. This was already observed in Fig. 5. Additional results on the variation of the number of latent dimensions is provided in Appendix C.

**Prediction of Synthetic Waxman: Metric and Results.** For the synthetic data with a single start-target pair, two metrics are evaluated: the Jensen-Shannon divergence ($D_{\text{JS}}$, lower is better) between edge usage in 1,000 test versus predicted (KDE) samples, indicating the similarity of edge frequencies, and Spearman's rank correlation (higher is better) between the common paths in the inferred and actual set of paths to assess the alignment in frequency ranking. Each method is sampled five times to compute the mean and standard deviation. IO-LVM outperforms the baseline PO (details of this baseline below) in Spearman's correlation, due to its ability to recover distinct costs in the unconstrained space, even in multimodal cases (e.g., three agents with different paths). In contrast, PO generates noisy paths around a (single) learned optimal set of transition costs, which may not align with the true distribution. The VAE, despite good training performance, failed to reconstruct valid paths, indicating poor generalization.

*Table 2.* Results on the prediction of paths distribution. NC denotes no convergence.

| Method | Synthetic | | Ship |
|--------|-----------|---|------|
| | $D_{\text{JS}}$ | Spearman | $D_{\text{JS}}$ sample |
| PO | $0.058 \pm 0.008$ | $0.813 \pm 0.025$ | $0.500 \pm 0.161$ |
| VAE | $0.112 \pm 0.002$ | $0.639 \pm 0.144$ | NC |
| IO-LVM | $0.056 \pm 0.003$ | $0.873 \pm 0.016$ | $0.467 \pm 0.195$ |

**Prediction of Ship Dataset: Metric and Results.** In the ship dataset, paths include multiple start and end nodes, making it infeasible to measure distribution distances for fixed start-target pairs due to the limited number (or even a single) of available paths per pair. Therefore, $D_{\text{JS}}$ is measured between the edges of each inferred sample and its corresponding test sample, and the average is computed across the dataset. For this evaluation, the most likely path from each model and baseline is compared to the observed paths. IO-LVM slightly better than PO, but the difference is not statistically significant due to high variance in the error metric. The VAE baseline failed to converge, likely due to the graph size and the complexity of having multiple start and target node scenarios.

**Baseline PO** Perturbed Optimizer (PO) Berthet et al. (2020) focus on recovering structured outputs also with gradient estimates, but without a latent space model. We adapt their method in two ways: (1) we learn based on paths without considering context, as the original paper is context-based and ours is not; and (2) to promote distribution reconstruction, we re-introduce the noise $\epsilon$ during inference.

## 4.5. Effect of varying $\beta$: Denoising versus Reconstruction

We analyze the effect of varying $\beta$ on three metrics in the synthetic Waxman dataset: i) the number of distinct paths reconstructed by the decoder using the test dataset; ii) the Fenchel-Young loss; iii) and the Intersection over Union (IoU) metric between observed and inferred edges usage during training. Table 3 shows that as $\beta$ increases, the number of distinct paths decreases, indicating a denoising effect due to the diminished influence of the reconstruction loss. This results in the decoder reducing diversity of generated paths due to the posterior collapse. The Fenchel-Young loss increases and the IoU decreases with larger $\beta$, also reflecting a reduction in reconstruction accuracy. An illustration of the correlation between Fenchel-Young loss and IoU is also observed in the learning curve (Fig. 9 in Appendix).

*Table 3.* Effect of varying $\beta$ on path reconstruction. Lower $\beta$ yields more distinct paths, while a balanced $\beta$ enables denoising. Higher $\beta$ leads to posterior collapse. FY denotes Fenchel-Young.

| $\beta$ | Distinct paths | FY train loss | IoU train |
|---------|---------------|---------------|-----------|
| 0.002 | 66 | 0.021 | 0.973 |
| 0.02 | 59 | 0.022 | 0.981 |
| 0.1 | 51 | 0.027 | 0.975 |
| 1 | 15 | 0.049 | 0.940 |
| 10 | 4 | 0.099 | 0.832 |
| 20 | 1 | 0.150 | 0.491 |

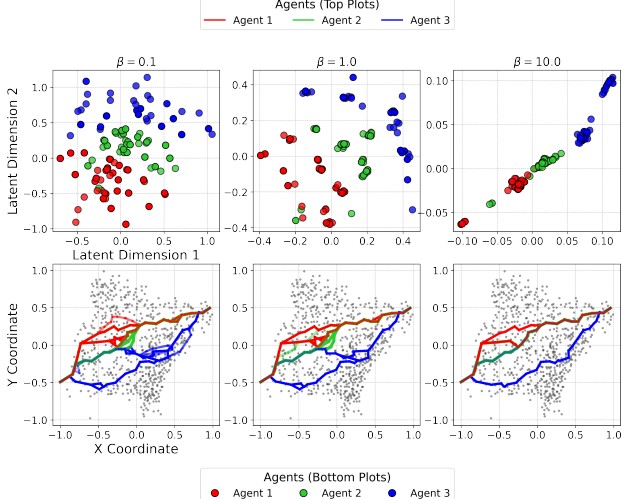

*Figure 6.* Varying $\beta$ in the latent space projection (top graphs) and in the reconstruction (bottom graphs).

## 5. Conclusion

This paper proposed IO-LVM, a novel approach for learning latent representations of COP costs, specifically for paths and cycles in graphs. The method leverages amortized inference and integrates black-box solvers within a probabilistic framework, allowing for the modeling of multiple agents and diverse behaviors in graphs. By employing a Fenchel-Young loss with perturbed inputs, it overcomes the gradient challenges in optimizing COPs, ensuring feasible and interpretable path reconstructions. The learned latent space captures meaningful structures, highlighting the model's characteristic to distinct agent behaviors, while maintaining accurate path reconstruction and prediction. Our method description is valid for a general set of COPs if gradient estimation is available.

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

## A. Datasets in more detail

**Synthetic Waxman Random Graph**   We generate a Waxman graph (Van Mieghem, 2001) with 700 nodes ($\alpha = 0.05$, $\beta = 0.6$), where the probability of an edge between two nodes $u$ and $v$ is given by $P(u, v) = \alpha \cdot \exp\left(-\frac{d(u,v)}{\beta \cdot d_{\max}}\right)$, where we considered $d(u, v)$ as the Euclidean distance between nodes $u$ and $v$, and $d_{\max}$ is the maximum distance between of two nodes, consequently ending up in 7230 edges. We create three edge cost sets to simulate three different agents performing decisions to go from start and end nodes. The edge costs are based on Euclidean distances, with higher costs for the southern edges for agent 1, and higher costs in the northern for agent 3, while agent 2 is not biased by the edges position. For each agent, we add a random noise in the cost elements $\mathbf{y}$ so the generated paths can be different from each other even within the same agent. The "observed" paths are generated by running the Dijkstra on the noisy edge costs. Two sets of 6,000 observed paths are generated: one with a single source and target pair (Fig. 2, top-left) and another with multiple source-target pairs (Fig. 2, bottom-left). In each of these sets, 5,000 paths are used for training IO-LVM and the baselines, while 1,000 are used for evaluation purposes. Further details on cost generation are provided in the code.

**Ships dataset**   We use the Automatic Identification System (AIS) data provided by the Danish Maritime Authority (Danish Maritime Authority, 2020), considering latitude and longitude projected in a 2D space for simplicity. The analysis focuses on paths from the first week of the months January 2024, May 2024, and June 2024. Only paths that exceed a distance of 4 units (in latitude/longitude) in Euclidean space are included. A path is considered completed either when the ship speed approaches zero or when there is an abrupt change in its heading. In some cases, there are gaps in the latitude/longitude signals; when such jumps occur, we segment the data and treat them as separate paths. We created a grid graph with a distance of 0.09 units between adjacent nodes, focusing on the area where there are more route options to be taken, which in total led to 2513 nodes and 8924 edges. This resulted in approximately 2,500 ship paths.

**TSPLIB**   We use datasets from TSPLIB95 (Reinelt, 1991), a library of benchmark instances for the Traveling Salesman Problem (TSP) and related optimization problems. Specifically, we selected two graphs: *burma14*, which consists of 14 nodes representing locations in Myanmar, forming a complete graph with 91 edges, and *bayg29*, which consists of 29 nodes representing the coordinates of cities in Bavaria, Germany, forming a complete graph with 406 edges. To assign the actual edge costs $\mathbf{y}$, we uses the Euclidean distance between nodes as an offset, and design a nonlinear function that incorporates unobserved features to calculate edge costs. We generate two datasets for each graph, one considering 3 unobserved features (less complex), and one considering 50 unobserved features (more complex). The observed paths are generated as $\omega(\mathbf{y})$ without noise, where $\omega$ represents a TSP search solver.

## B. Implementation details

IO-LVM is trained according to Algorithm 1. We used PyTorch (Paszke et al., 2019) for the implementation. We consider a learning rate of 0.00004 and a batch size of 250 for the Waxman synthetic dataset and the Ship dataset, while the Hamiltonian Cycle experiment uses a learning rate of 0.0001 and a batch size of 200. COPs are solved in parallel for the batches. The neural network architectures do not have any special implementations. The encoder architecture consists of a neural network with 4 hidden layers, each containing 1000 neurons, with ReLU activation functions in the hidden layers. The decoder architecture is the same as the encoder, but with a Softplus activation function to ensure that all edge costs remain positive. The RMSProp optimizer is used for the Synthetic and Ship datasets, while the AdamW optimizer is used for the Hamiltonian Cycles experiment. The experiments were run on a CPU due to the bottleneck introduced by COP solvers. The processor model used was the 13th Gen Intel(R) Core(TM) i7-13700KF, which has 24 cores. We use Dijkstra from networkx library in python (Hagberg et al., 2008) for solving SPP: and Ortools routing python library for TSP solutions (Furnon & Perron). Further details are in the provided code.

## C. Varying the number of latent dimensions

In our experiments, we observed that for certain tasks, a very low number of latent dimensions was enough. For instance, in the Ship Dataset, attempting to add a third dimension to the latent space revealed that the second and third dimensions are highly correlated (Fig. 7), indicating that the third dimension is unnecessary.

Conversely, in the Hamiltonian Cycles experiment, where 50 hidden features were used to generate edge costs with a complex relationship, increasing the latent dimensions proved beneficial in mitigating underfitting during the reconstruction process. This effect is illustrated in Fig. 8, which compares the performance of using 2 latent dimensions (left graphs) versus 10 latent dimensions (right graphs) for both the *burma14* (top graphs) and *bayg29* datasets. In the right-hand charts, we observe that using 10 latent dimensions achieves 100% Recall in the reconstructions for the training datasets and improves Recall for the validation datasets compared to the 2-dimensional case shown in the left-hand charts. However, with 10 latent dimensions, a

slight overfitting emerges, which could be mitigated through more careful regularization and neural network architecture design. Addressing this was beyond the scope of our current study.

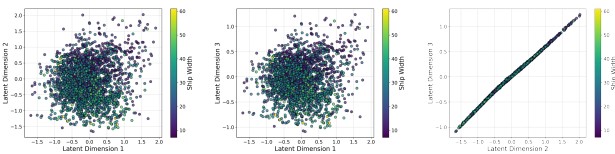

*Figure 7.* Latent space of ship trajectories using three dimensions. The right graph indicates that there is no need for a third latent dimension. Narrow ships are more concentrated in the top right corner of the two left graphs.

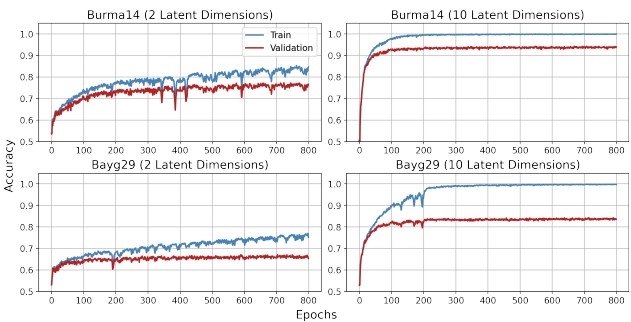

*Figure 8.* Comparison of training and validation curves for reconstruction performance using 2 latent dims (left) and 10 (right) for the *burma14* (top) and *bayg29* (bottom) datasets. Increasing the number of latent dims improves Recall for both training and validation datasets.

## D. Additional Figures

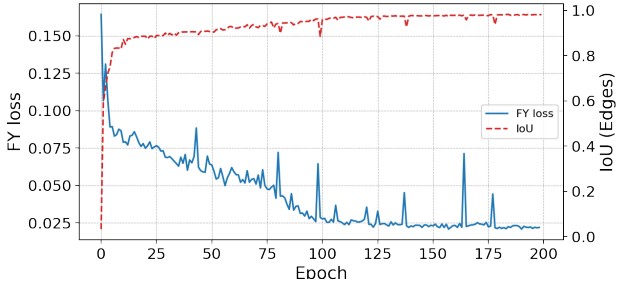

*Figure 9.* Fenchel-Young loss and IoU between edges computed during the training process.

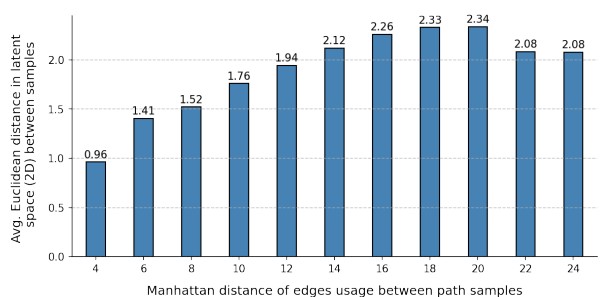

*Figure 10.* Bar plot showing the relationship between Manhattan distance groups (horizontal axis) and the average Euclidean distance in the latent space (vertical axis). The plot illustrates that samples with smaller Manhattan distances (i.e., paths with more similar edge usage) tend to have smaller Euclidean distances in the latent space.

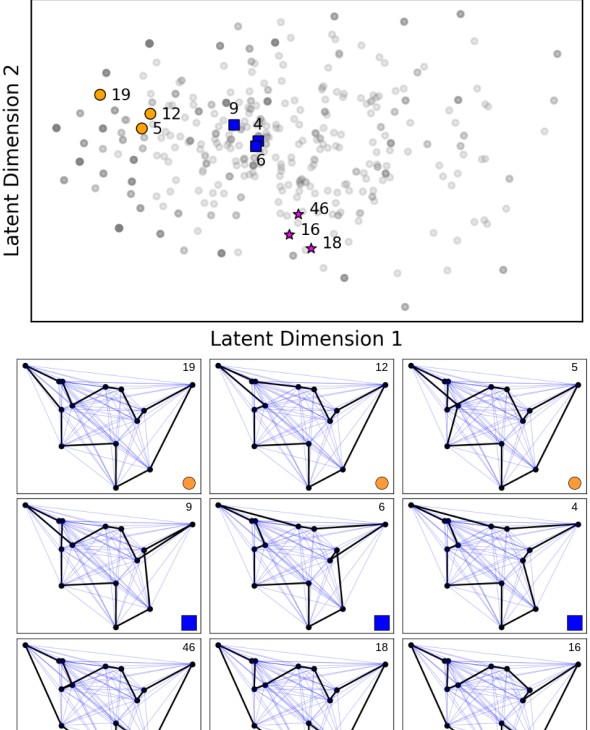

*Figure 11.* Visualization of the latent space and paths for the *burma14* graph. The top graph shows the latent space with nine manually selected samples. The bottom graphs display the corresponding paths for these samples.

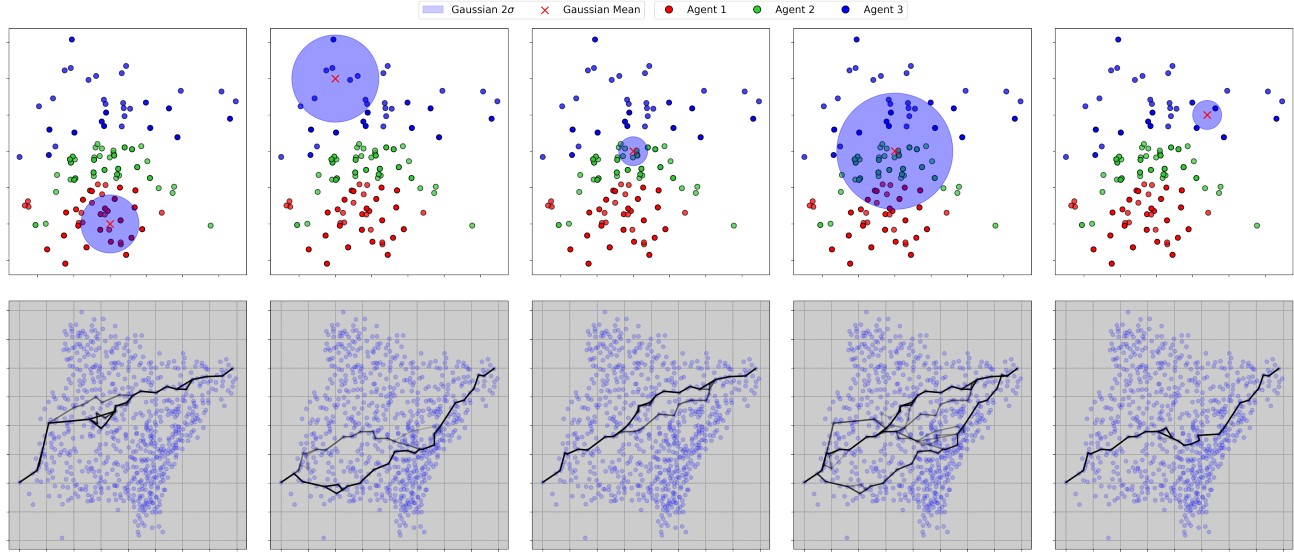

*Figure 12.* Reconstruction for the synthetic data with single pair of start and end nodes. Top charts: region of samples from a Gaussian in the latent space. Bottom charts: corresponding generated trajectories. Blue agents has higher costs on edges in the north, while red edges has higher costs on edges in the south.

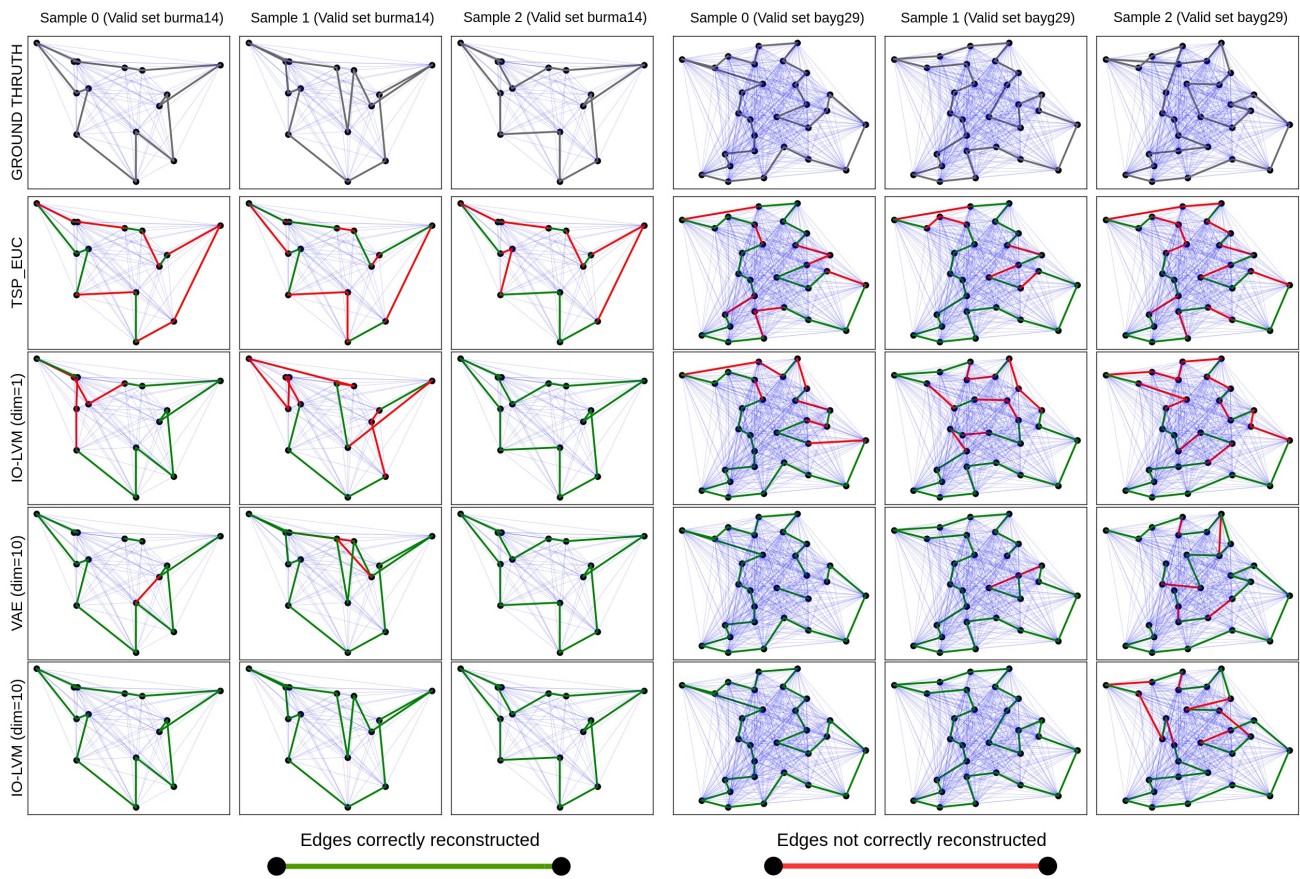

*Figure 13.* Each column illustrates an inferred sample for the Hamiltonian Cycles experiment. The first row represents the cycle observed in the validation set. The second row represents a solution of the TSP using edges cost as euclidean distances (offset in the data generation process). Third and Fourth rows represent inference using VAE and IO-LVM with 10 latent dimensions. Green edges denote correct reconstructions relative to the groundtruth, while red edges indicate false positives.

