# OpenReview forum: "IO-LVM: Inverse Optimization Latent Variable Models with Graph-based Planning Applications"
_ICML.cc/2025/Conference — Submitted to ICML 2025_

### Official Review · Reviewer_Xit2 · 2025-03-08

**Overall Recommendation:** 2

**Summary:**

This paper introduces IO-LVM (Inverse Optimization Latent Variable Models), and this method learns latent representations of constrained optimization problem (COP) costs based on observed solutions, with applications to graph-based planning problems. The authors come up with a new variate of VAE training loss so that the cost vector can be inferred and serve as the input of the solver. They conduct experiments on graph data in different scenarios.

**Claims And Evidence:**

From the experimental results, the method works well in path planning and other tasks. I really like the analysis of latent space in this work. However, in Table 1, I feel the improvement of the proposed method is marginal, comparing the vanilla VAE baseline. Also, I think the author should add more baselines, and I'm wondering how IO-LVM works comparing the RL-based method on graph data.

Can the author generalize the method to other modalities? Since the transformation from z to y is very flexible, it should be easy to demonstrate it to other datasets besides graph optimization. Also, the Hamiltonian Cycles results are kind of saturated, so it is hard to evaluate the potential of the method.

**Essential References Not Discussed:**

N/A

**Experimental Designs Or Analyses:**

The experimental designs make sense to me. It would be great if the author could provide more complex datasets/modalities and stronger baselines in this work.

**Methods And Evaluation Criteria:**

The idea of this method really makes sense to me. But again, I feel the authors can provide more baseline methods. Also, I'm curious how the VAE baseline is trained here. Does it also include a solver, and how is the y inferred in this case?

For the optimization method, people may use RL-based models. How does IO-LVM compare to RL methods?

**Other Comments Or Suggestions:**

N/A

**Other Strengths And Weaknesses:**

The presentation of the paper is clear and easy to read. The training strategy has novelty. It should be scaled up and expanded to other domains. However, the experimental results are not strong and convincing enough.

**Questions For Authors:**

See the questions above.

**Relation To Broader Scientific Literature:**

N/A

**Theoretical Claims:**

There's no theoretical issue in this paper.

---

> ### Author Rebuttal · Authors · 2025-03-31
>
> The main points from the reviewer are:
> * Q1) How the baseline VAE is trained to output y
> * Q2) Marginal improvement only compared to VAEs (Table 1)
> * Q3) RL-based and other baselines (also to reviewer 2TrQ)
> * Q4) Problems other than graphs
>
> .
>
> * Answer to Q1 and Q2.
>
> This is a key question. Actually, the VAEs do not output Y, they output X directly (i.e., the paths). There is no way for VAEs to output Y because we do not have training data for Y. Our method, instead, outputs first Y (even without their labels), and then input Y to the solver to finally reconstruct X (i.e., the paths), assuring its feasibility with respect to the solver/COP. For this reason, while VAEs are not capable to guarantee the reconstruction of a feasible path, our method always reconstructs a feasible path, even if it does not reconstruct the path correctly. Please see the additional illustration: https://docs.google.com/document/d/e/2PACX-1vTI3F6-L4XLqFv9NWiE9LriBkZ-CpIAhvtd9XGDyQr59PAQBv-IG_z8-EaqIQlBXlZwxqeCJNgtqjxA/pub
>
> When the reviewer ask about the result of Table 1, at the first moment it seems a small improvement. However, we were only reporting the reconstruction of edges reconstruction. In many cases, VAEs might capture e.g., 95% of the edges correctly, but when you put those edges together, they do not form a Hamilt. cycle.
>
> In order to complement our answer, we provide three more tables (find below). The first table shows the % of Hamilt. cycles that are fully correctly reconstructed (i.e., compare the full vector and mark it as correct if it matches perfectly). Note that this is a much harder task than the % of edges correctly reconstructed. The second table shows the percentage of outpus that are Hamilt. cycles (even if they are not correctly reconstructed). The third table shows the reconstruction results given different data training size (i.e., how much data is needed so that VAE versus IO-LVM can reconstruct the correct Hamiltonian cycle?)
>
> **TABLE 1 (IO-LVM versus VAE: How many full Hamiltonian cycles are correctly reconstructed? (Avg and std over 5 runs))**
> | Methods    | Latent Dims | burma14 (3 dims) | bayg29 (3 dims) | burma14 (50 dims) | bayg29 (50 dims) |
> | -------- | ------- | ------- | ------- | ------- | ------- |
> | VAE  | 2  | $55.2 \pm 1.1 $ \% | $15.4 \pm 1.3 $ \% | $9.8 \pm 1.3 $ \% | $0.2 \pm 0.1 $ \%
> | IO-LVM | 2  | $78.4 \pm 0.6$ \% | $37.3 \pm 7.3$ \% | $29.6 \pm 2.6 $ \% | $1.5 \pm 0.2 $ \%
> | VAE  | 10   | $ 82.7 \pm 1.3 $ \% | $46.0 \pm 1.6$ \% | $45.3 \pm 1.9 $ \% | $3.4 \pm 0.5 $ \%
> | IO-LVM  | 10   | $ 91.0 \pm 0.7 $ \% | $77.1 \pm 1.5$ \% | $78.3 \pm 1.1 $ \% | $31.9 \pm 3.5 $ \%
>
> .
>
> **TABLE 2 (IO-LVM versus VAE: What is the percentage of feasible outputs (even if they are not correctly reconstructed)? Note that all IO-LVM outputs are feasible (100%) due to the solver in the loop, enhancing the results in the first table. (Avg and std over 5 runs))**
> | Methods    | Latent Dims | burma14 (3 dims) | bayg29 (3 dims) | burma14 (50 dims) | bayg29 (50 dims) |
> | -------- | ------- | ------- | ------- | ------- | ------- |
> | VAE  | 2  | $63.2 \pm 1.6 $ \% | $21.3 \pm 3.1 $ \% | $18.3 \pm 2.5 $ \% | $0.5 \pm 0.3 $ \%
> | IO-LVM | 2  | $ 100  \pm 0.0 $ \% | $100  \pm 0.0$ \% | $100  \pm 0.0 $ \% | $100  \pm 0.0 $ \%
> | VAE  | 10   | $ 85.0 \pm 1.4 $ \% | $50.1 \pm 1.6$ \% | $47.5 \pm 2.2 $ \% | $4.5 \pm 0.7 $ \%
> | IO-LVM  | 10   | $ 100  \pm 0.0 $ \% | $100  \pm 0.0$ \% | $100  \pm 0.0$ \% | $100  \pm 0.0$ \%
>
>
>
> .
>
> **TABLE 3 (How much training data does IO-LVM versus VAEs need to reconstruct the Hamiltonian cycles correctly? Here we fix the seed of the model.)**
>
> | Methods    | Latent Dims | Train size = 500 | = 1000 | = 1600 |  = 2400 |  = 4000 | = 7000 | = 10000 |
> | -------- | ------- | -------- | ------- | -------- | ------- |  ------- | ------- |  ------- |
> | VAE  | 10 | $0.0$ \%| $0.8$ \%| $3.3$ \%| $3.9$ \%| $6.2$ \%| $11.5$ \% | $11.8$ \% |
> | IO-LVM | 10 | $4.8$ \%| $20.8$ \%| $30.5$ \%| $33.0$ \%| $42.5$ \%| $53.5$ \%| $58.3$ \%|
>
>
>
> * Answer to Q3.
>
> We don't have space left here, so we decided to answer questions regarding related work in the rebuttal for reviewer sTUc (Q3)
>
>
>
> .
>
> * Answer to Q4.
>
> Yes! The reviewer is correct in observing that the method could be scaled to problems other than graph-based problems, also because "the transformation from z to y is very flexible" . We decided to provide experiments in graph-based applications because it helps us to dive deep into the interpretability/qualitative world. In other words, we aimed to write a paper that not only rely on quantitative results on reconstructions and predictions (as we believe this is saturated in many ML subfields), but most importantly with illustrations on how latent dimensions, inputs and reconstructions are connected, and how non-observed factors are also captured in the latent space, which would be less effective to show with applications other than graphs due to its nice visual representations.

---

### Official Review · Reviewer_NTyH · 2025-03-12

**Overall Recommendation:** 3

**Summary:**

This paper develops a latent variable model for constrained optimization problems like path planning in graphs. The key insight is to modify the traditional Variational Auto-Encoder (VAE) with two distinct stages of reconstruction - from the latent space to an unconstrained space, and from the unconstrained space to the constrained space.

Experiments through path planning simulations reveal that the proposed approach (IO-LVM) captures the dataset's path distributions and enables interpretable predictions via latent space compression.

**Claims And Evidence:**

The paper makes several claims:

1) IO-LVM naturally constructs a disentangled, and sometimes multimodal, latent space, allowing for the reconstruction of observed path distributions without making assumptions about inferred paths

2) IO-LVM learns interpretable latent representations

3) IO-LVM allows for predicting how different agents might navigate between unseen source and target nodes, providing a flexible framework for path inference.

Claims 1) and 2) are not well substnatiated - there are no results showing that the latent representations learned are distengled / interpretable compared to standard VAEs

Claim 3) is really substantiated through path planning simulation experiments.

**Essential References Not Discussed:**

None to my knowledge

**Experimental Designs Or Analyses:**

The experiments seem sound and valid, but as mentioned in the previous comment - the scope of the experiments are fairly limited. It is unclear in which real-world constrained optimization problems the proposed approach might be preferred compared to more standard VAE-based solutions and other constrained optimization solutions that do not model the problem as a latent variable model.

Additionally, the experiments do not really provide insights into the interpretability / disentanglement of latents, which is one of the key claims of the paper.

**Methods And Evaluation Criteria:**

The scope and experimental evaluations of the paper are fairly limited: the paper seems to target general constrained planning problems and motivates the model based on this, but the experiments only focus on simple 2D simulations of path planning in graphs.

**Other Comments Or Suggestions:**

The paper is not well-written, with confusing texts, and a lot of jargon.

For example the abstract starts with " (Variational) Autoencoders struggle to capture constraints to decode structured outputs." - it is unclear what is meant by structured outputs here.

I would ask the authors to revise the writing of the paper to make it more accessible, crisp, and clearly isolate the claims and contributions.

**Other Strengths And Weaknesses:**

Strengths

- the paper proposes a novel modification to a VAE by changing the reconstruction to consist of two stages.

- the paper is well-motivated from the perspective of a latent variable model i.e. to promote learning interpretable and compressed latent representations, such that the data can be faithfully reconstructed.

- the paper has many experiments on simulated 2D graphs that demonstrates an implementation of the proposed model, and its benefits compared to other latent variable models.


Weaknesses

- the claims of the paper around interpretability are not well-substantiated

- the experiments are on toy settings of 2D graphs and thus it is unclear how widely applicable are the benefits from the proposed approach

- given the scope of the experiments, it is unclear how impactful the proposed LVM will be compared to current models in adoption for real world applications like robotics, and path planning.

**Questions For Authors:**

Please refer to the weakness and address them to the extent possible.


- the claims of the paper around interpretability are not well-substantiated

- the experiments are on toy settings of 2D graphs and thus it is unclear how widely applicable are the benefits from the proposed approach

- given the scope of the experiments, it is unclear how impactful the proposed LVM will be compared to current models in adoption for real world applications like robotics, and path planning.

**Relation To Broader Scientific Literature:**

The paper broadly relates to literature on constrained optimization, latent variable models, and path planning problems in graphs.

**Theoretical Claims:**

There are no proofs or theoretical claims.

---

> ### Author Rebuttal · Authors · 2025-03-31
>
> The main points from the reviewer are:
> * Q1) Results showing that the latent representations learned are distengled / interpretable compared to standard VAEs
> * Q2) Concern of the graphs in the experiments being only in 2D
> * Q3) Writing
> * Q4) In what type of applications the proposed approach might be preferred compared to more standard VAE-based
>
> .
>
> * Answer to Q1.
>
> We thank the reviewer for the question. We want to highlight that we do not claim that our latent representation is "more interpretable" than VAEs, we claim that we maintain the disantanglement/interpretability of VAEs, but increase the reconstruction power by guaranteeing constraint feasibility through a suitable solver, such as Dijsktra and TSP solver, in a full differentiable framework.
>
> How does our LVM provides interpretability?
>
> Although it is hard to "quantify" interpretability, the book [Probabilistic Machine Learning, Kevin Murphy] states that "Interpretability allows human expert to inspect a model ... understainding why they work to gain scientific and operational insight". In our paper, for example, Figures 11 and 12 provide insight on the latent space (e.g., how a walk through the latent space impacts the encoding and reconstruction). In Figure 3, we highlight the correlation of an unseen feature with the latent values, giving the possibility for an expert to inspect if the model is capturing the hidden context correctly.
>
> Why does our LVM also have the disantanglement property as VAEs and how do we show that?
>
> From the paper [Disentangling Disentanglement in Variational Autoencoders, 2019], disantanglement makes each latent dimension to control one interpretable attribute. This is observed, for example, in Figures 2 and 3 in our paper. More specifically, Figure 3 highlights that the latent dimension in the vertical axis is correlated to the "ship width" feature. Note again that the "ship width" was not even observed in the data, but was an underlying important factor to make ships to perform different paths.
>
> Compared to a normal VAE, we additionally know that the latent space needs to capture the costs (solver input) and not the shape of the solutions (solver output).
>
> Please, see an extra illustration that we provide in the following link:
> https://docs.google.com/document/d/e/2PACX-1vTI3F6-L4XLqFv9NWiE9LriBkZ-CpIAhvtd9XGDyQr59PAQBv-IG_z8-EaqIQlBXlZwxqeCJNgtqjxA/pub
>
> .
>
> * Answer to Q2.
>
> In our framework, there is no restriction regarding the graph being in 2D. Note, for example, that in the Start/target path problem, Dijsktra solver is used, which is a dimension-agnostic algorithm. In the Hamiltonian cycle problem, note that a TSP solver is used without leveraging 2D heuristics. In Figure 5, for example, considering the Sample 2 of "Bayg29", it is clear that there are edges crossing in the solution. It would not be possible to output such a solution with a solver with 2D restrictions, since in 2D the optimal TSP solution has no edges crossing. We do not leverage any benefit of the graph to be in 2D other than for visualization purpose.
>
> Furthermore, we emphasise that, if we are dealing with an NP hard problem such as the TSP, it is natural that the problem size is smaller.
> In previous works published in important venues such as [Learning with Differentiable Perturbed Optimizers, 2020], [Differentiation of Blackbox Combinatorial Solvers, 2019] and [DataSP: A Differential All-to-All Shortest Path Algorithm, 2024], they deal with graph problems with approximately the same (or smaller) size than ours.
>
> .
>
> * Answer to Q3.
>
> Since other reviewers did not point out to writing issues, and the reviewer provided a specific example, we answer this example as follows. The term "structured output" can be found many times, for example, in the book [BakIr, G. (Ed.). (2007). Predicting structured data]. The term "structured output" is generally related to a model or a function yielding a structured vector where its elements has interdependency, such as a sequence, a path, or a decision that should adhere to specific constraints. We are adding this reference in the introduction of the paper to proper relate the term used to the literature.
>
> .
>
> * Answer to Q4.
>
> We have added extra results and analysis to clarify more the power of IO-LVM in comparison to standard VAEs. Please see the new tables in the answer to reviewer Xit2. There we show three extra results: i) When we compare the full output/reconstruction instead of comparing the edges reconstruction, the IO-LVM is much better than VAEs. ii) We show that generally when VAEs make reconstruction mistakes, the mistakes are non-structured (e.g., the reconstruction is not a Hamiltonian cycle), which is not true for IO-LVM, that even when making reconstruction mistakes, it guarantees COP feasibility. iii) We show that IO-LVM needs much less training data to give reasonable reconstruction results compared to VAEs.

---

> > ### Comment · Reviewer_NTyH · 2025-04-04
> >
> > thank you for the clarifications! Since the authors answered my questions around the claims, and provided more results on other domains, I am increasing the score. Although I am still not convinced by the results (the new results seem to be again on simple 2D settings which may not translate to the diversity of complex real-world scenarios across problems in planning, robotics etc. that standard VAEs have proven useful) - as such I will not fight for acceptance.

---

> > > ### Author Response · Authors · 2025-04-04
> > >
> > > Thank you again for your feedback and to engage with the rebuttal.
> > >
> > > It is important to emphasise that our framework is not dependent on the graph dimensionality. Our method inherently operates on graph-based structures, treating inputs as graph entities composed of nodes and edges. It relies solely on the topology of the graph. For instance, in our Hamiltonian cycles experiments, the input graph is fully connected, which represents a dense, high-complexity scenario regardless of any geometric embedding. If we have to be very picky, in all graphs that are related to latitude and longitude points, they are actually in 3D due to the earth's curved surface.
> > >
> > > With all respect, it is hard to agree that our paper is under simpler problem settings when we see related works of combining deep learning and combinatorial solvers applying to smaller or similar size of graph problems. See the examples below of well established papers in the area:
> > >
> > > * For the start/target planning problem, in the work [Differentiation of blackbox combinatorial solvers (2020)], the authors deal with graphs between 144 and 900 nodes (see table 2 in the mentioned paper) and with a maximum of around 2000 edges, while we deal with graphs between 700 and 2513 nodes, reaching up to 8924 edges. For the Hamiltonian cycle problem, they vary the number of nodes from 5 to 40 (see table 3 in the mentioned paper), while we vary from 14 to 29 (our method can also go up to 40 or more).
> > >
> > > * For the start/target planning problem, in the work [Learning with Differentiable Perturbed Optimizers (2020)], the authors deal with graph problems of size 144 (see subsection 5.3 in the mentioned paper), while we deal with graphs between 700 and 2513 nodes.
> > >
> > >
> > > Although we acknowledge that the scalability of VAEs is higher, they can't guarantee feasible outputs, which is the key point of our method (see Table 2 in the answer for reviewer Xit2).

---

### Official Review · Reviewer_sTUc · 2025-03-18

**Overall Recommendation:** 3

**Summary:**

The paper proposes a representation learning based method for generating feasible solutions for constrained optimization problems. Given a black-box solver and a dataset of constraints and their solutions, the proposed IO-LVM algorithm learned a latent representation model which is used to generate feasible solutions by sampling in the latent space. Experiments of path planning and Hamiltonian cycles problems are shown the demonstrate the generated results of IO-LVM.

**Claims And Evidence:**

Since the reconstruction loss in ELBO is non differentiable, the paper proposes to replace it with the Fenchel-Young loss and derive an estimator for the corresponding gradients. The of using this loss makes intuitive sense. However, no theoretical evidence supporting if minimizing this loss can lead to a good latent representation.

The efficacy of IO-LVM is mainly supported by the numerical experiments. For the latent representation, Figure 2-4 and some qualitative properties are provided to support that the latent space does learn something. For quantitative analysis, two kinds of tasks are considered. One is the capable to reconstruct the Hamiltonian cycles, and another one is the quality of the generated paths based on edge usage compared with the testing set.

**Essential References Not Discussed:**

None.

**Experimental Designs Or Analyses:**

The experimental designs make sense and analysis from multiple angles are provided.

**Methods And Evaluation Criteria:**

The proposed IO-LVM method and evaluation criteria mentioned in the previous section make sense for the problem.

One key restriction of the proposed method is the requirement of a black box solver, but little is discussed on how the solver would be available.

**Other Comments Or Suggestions:**

No additional comments.

**Other Strengths And Weaknesses:**

Using a latent representation learning approach to find feasible solution for constrained optimization seems to be a novel and reasonable idea. The numerical results show some good performance of the proposed IO-LVM.

As mentioned that the requirement of the black-box solver is a key limitation. Given that IO-LVM has access to the solver, the comparisons with baselines could be a bit unfair as the baseline methods don't make use of the solver.

**Questions For Authors:**

Can you provide more justification of the availability of the black-box solver?
Are there baselines using the solver that can be compared in a fair manner?

**Relation To Broader Scientific Literature:**

The paper provides examples of applying IO-LVM to path planning and Hamiltonian cycles, and these two types of problems have applications in various scientific areas.

**Theoretical Claims:**

No theoretical claims.

---

> ### Author Rebuttal · Authors · 2025-03-31
>
> The main points from the reviewer are:
> * Q1) Requirement of a blackbox solver in the framework.
> * Q2) No evidence supporting if minimizing the fenchel yong loss can lead to a good latent representation.
> * Q3) Baselines / Related work
>
> .
>
> * Answer to Q1.
>
> The requirement of a blackbox solver in the learning process is present in many topics such as “Decision-focused” learning, “Predict-and-Optimize”, and some subtopics in Imitation Learning as well such as [Task-based End-to-end Model Learning in Stochastic Optimization, 2017]; [Differentiation of Blackbox Combinatorial Solvers, 2019]; [Learning with Differentiable Perturbed Optimizers, 2020]. One of the key points in all of those papers, including ours, is to leverage years of knowledge of Mathematical Programming into Deep Learning methods. For example, Dijkstra solves efficiently the Shortest Path Problem, so our method learns the cost inputs (edges costs) of Dijkstra, and we will be sure that the output is feasible wrt constraints of the problem. Our solution is then more general than other approaches as we do not need to handle the constraints of a particular problem. Note that we are using the blackbox solver to recover the costs that generated those observations.
> The only baseline we use that does not leverage the solver is the VAE / $\beta$-VAE. But since we are proposing a generative model here, we believe it is natural to compare to standard VAEs to show the benefit of constraining the output through the solver.
> Indeed, we have used the last above-mentioned work as our baseline for path distribution prediction (Table 2 in the paper), which also leverages blackbox solver in the loop. However, it is not possible to use the same baseline into the reconstruction experiment because they do not model a latent space.
>
>
>
> .
>
> * Answer to Q2.
>
> We understand the reviewer’s concern regarding the justification for the Fenchel-Young loss leading to a good latent representation. In standard VAE literature, it is well-known that the interplay between a reconstruction loss and a regularization term (typically the KL-divergence between the posterior and prior distributions) promotes interpretability, smoothness, and disentanglement of the latent space. However, rigorous proofs of these properties often depend on restrictive and sometimes unrealistic assumptions.
> Other studies show empirically that Beta-VAE variants affects (sometimes in a good way) the interpretability and structure of latent representations by simply adjusting the hyperparameter Beta (trade-off between reconstruction and regularization). However, providing a theoretical explanation of such effects remains challenging within deep learning.
> In our approach, we reiterate the need of both: a reconstruction loss and a regularization term. Given that COP solvers are inherently non-differentiable, we rely on gradient estimations through perturbed COP solutions provided by blackbox solvers. According to [Learning with Differentiable Pert. Optimizers, 2020], the Fenchel-Yong loss is convex with respect to the parameters $\theta$ and attains a minimum (zero) when the predicted structure matches the ground truth structure exactly. This motivates our choice of the Fenchel-Young loss as a suitable (and demonstrated to be effective) reconstruction loss.
>
> Please see: https://docs.google.com/document/d/e/2PACX-1vTI3F6-L4XLqFv9NWiE9LriBkZ-CpIAhvtd9XGDyQr59PAQBv-IG_z8-EaqIQlBXlZwxqeCJNgtqjxA/pub
>
>
>
>
> .
>
> * Answer to Q3.
>
> We use this space to answer questions to reviewers 2TrQ, sTUc and Xit2.
>
> We thank the reviewers for highlighting (Inverse) RL-based methods and GFlowNet (2023). Although relevant, these methods require explicit encoding of hard constraints in state transition logic, introducing manual modeling efforts specific to each problem. In contrast, IO-LVM inherently includes constraints via the solver, enabling easy adaptation by simply replacing the solver.
>
> The suggested [Scalable Equilibrium Sampling with Seq. Boltzm. Generators (2025)] employs probabilistic constraints through energy landscapes that may occasionally allow violations, requiring post-processing in the inference. Our approach does not have this limitation due to the direct blackbox solver integration.
>
> Traditional inverse RL methods (e.g., [MaxEnt inverse reinforcement learning, 2008] and related works) lack latent variable modeling, limiting them to a single recovered cost function. For example, Figures 2 and 3 in our paper illustrate distinct learned cost functions spread into the latent space, which would not be possible with more traditional Inverse RL methods .
>
> We are updating our references accordingly and adding a dedicated paragraph in our Related Work section to explicitly highlight these differences between solver constraints guarantee (more flexible) and manually designed action space or the need of post processing for constraints guarantee (less flexible). We thank the reviewers to provide constructive feedback in this regard.

---

### Official Review · Reviewer_2TrQ · 2025-03-22

**Overall Recommendation:** 3

**Summary:**

The paper provides a method to learn the representations of the underlying "cost" functions of trajectories in a structured domain. They achieve it by using a VAE like method that uses a "fenchel young" loss function instead of the reconstruction error. The representations are supposed to encode the cost that the input trajectory was optimizing for, given this representation, a classical solver solves for the optimal trajectory. The representations are learnt such that this reconstructed optimal trajectory should be close to the input trajectory.

**Claims And Evidence:**

I have a few questions about the loss functions, it could be possible that I am missing something:
1) How do you define the inner product between "y" an output of a neural network and "x" a structured domain like a graph?
2) Assuming the inner product is defined somehow, In equation 6, when estimating the gradient, why is \hat{x}_{\epsilon} treated as a constant with respect to \theta?
3) Alright, upon reading the experiment settings, it seems that in both problems (Path Planning and Hamiltonian Cycles) the domain of Y and X is the same. But what happens if it is not?

**Essential References Not Discussed:**

Mentioned above regarding boltzmann samplers, diffusion samplers, Gflownets, etc.

**Experimental Designs Or Analyses:**

Yes, I checked the both the qualitative and the quantitative results. The qualitative results make sense to me, epsecially the "unsupervised clustering" based on the true underlying cost. This is similar to unsupervised learning in CV, where images from the same classes have the similar VAE representations, despite access to labels. Hence these results are not very surprising to me. While the problems seems to be toyish problems, I am not a researcher in this area, hence I am unable to quantify the importance of the problems / result numbers.

Q) Why does the VAE only use 10 dimensional representation space?
Q) Why don't any of the numbers (in Table 1 for example) have error estimates?
Q) Aren't there any other benchmarks? This seems similar to boltzmann sampling with a non differentiable (and costly) reward function. There is a large amount of literature that deals with this, even with structured space more complex than the ones mentioned in the paper.
See for example:

1) GFlowNet Foundations.
2) Scalable Equilibrium Sampling with Sequential Boltzmann Generators (references in this paper).
3) Understanding reinforcement learning-based fine-tuning of diffusion models: A tutorial and review.
4) (Discrete) Diffusion samplers.

I wonder why these are not covered either in the related work section or as baselines.

**Methods And Evaluation Criteria:**

I am not experienced in this area, but it seems like all the problems are sort of toy / small scale problems by deep learning standards.

**Other Comments Or Suggestions:**

1) I don't think error plots in Table 1, and number of seeds in Table 1 and 2 are included.

**Other Strengths And Weaknesses:**

Please see strength and weaknesses above.

**Questions For Authors:**

I have mentioned my questions above.

**Relation To Broader Scientific Literature:**

The paper is well motivated, the problem of doing constrained generation / search in structured space is important. But in the current state, I personally do not think the papers findings contribute

**Theoretical Claims:**

There weren't a lot of theoretical details in the paper about the fenchel young loss, I do have a couple of clarifications which I have mentioned above.

---

> ### Author Rebuttal · Authors · 2025-03-31
>
> The main points from the reviewer are:
> * Q1) Y space restriction and inner product formulation.
> * Q2) Equation 6 (gradient estimation)
> * Q3) Limited-size problems compared to standard Deep Learning
> * Q4) VAE has low number of latent dimensions
> * Q5) Related work and baselines
> * Q6) Tables 1 and 2 (number of seeds and std)
>
> .
>
> * Answer Q1
>
> The reviewer has an important question about the Y space restriction. The restriction of Y space in our proposed framework comes from the requirement that Y space must align with the space of the COP cost vector. Hence, the reviewer's concern can be restated as: "Is it sufficiently expressive to consider only COP formulations with linear costs (inner products)?"
> The answer is yes. The way we formulate the problem, i.e., $argmin_x <y,x>$ (see Eq. 5) subject to a general non-convex constraint is highly expressive as many combinatorial problems can be translated into it.
> According to [Korte, Bernhard H., et al. Combinatorial optimization], “Virtually all combinatorial optimization problems can be formulated as integer programs…”, referring to integer linear programs in the book’s chapter.
> In fact, our formulation is even more general than integer linear programming (ILP) because we allow the constraints to be non-linear and non-convex.
> Therefore, it is a natural choice and expressive enough to choose Y to have the same number of dimensions of X. We will address this by adding a small text in section 3.2 between Equations 4 and 5 with the motivation above about the expressiveness of the linear cost formulation.
>
> .
>
>
> * Answer Q2
>
> We thank the reviewer for this observation, as we missed the $\epsilon$ term in the previous equations (Eq 2 and 5) that led to the confusion in Eq 6. To make sure the reviewers follow the equations, here are the steps considering the correction:
>
> Eq 2. $l_{\text{FY}}^{\epsilon}(y, x) = f(y, x) - f(y + \epsilon, \hat{x}_{\epsilon})$
>
> Eq 5. $E_{q_{\phi}(z | x)} \left[ \langle y^{\theta}, x \rangle -  \langle y^{\theta} + \epsilon, \hat{x}_{\epsilon}^{\theta} \rangle \right]$
>
> Now Eq. 6 follows correctly from Eq. 5. The gradient of the loss wrt $y$ in the first term is naturally $x$.
> The second term is $\hat{x}_{\epsilon}$, following Berthet et al. (2020) in [Learning with Differentiable Perturbed Optimizers] in Definition 2.1. Therefore Eq. 6 is correct.
>
> To give an intuition why this is true, by definition $\hat{x}_{\epsilon}=argmin_x [ \langle y + \epsilon, x \rangle]$, which is the gradient of the perturbed cost function $\langle y + \epsilon, x \rangle$, at the minimum value, where x = x_eps. Since the steps are not obvious, we are adding them to the appendix, additionally to the corrections above in the main text.
>
> .
>
> * Answer Q3
>
> Related works published in important venues such as [Learning with Differentiable Perturbed Optimizers, 2020], [Differentiation of Blackbox Combinatorial Solvers, 2019] and [DataSP: A Differential All-to-All Shortest Path Algorithm, 2024], tackle graph problems with comparable or smaller sizes than ours. There is a trade-off of using solvers within learning frameworks such as those above mentioned and ours. While these approaches guarantee the feasibility of COP solutions leveraging capabilities of blackbox solvers, they generally have lower scalability compared to standard deep learning architectures.
>
> .
>
> * Answer Q4
>
> This question is important to be discussed, and we would like to address it using Figure 2 in our paper. You can see that the bottom graph contains many different paths. Although the variaty of paths are high, they were generated using simple cost functions with noise (i.e., three cost functions + noise illustrated with different colors). Only few latent dimensions were enough to encode the cost information. We want to show that we are able to encode the underlying costs by inputing the paths (Inv. Optimization), while VAEs would need too much data to understand the inverse mapping. Fixing the number of latent dimensions, IO-LVM needs less data to capture the structure. We added this new result in the rebuttal to reviewer Xit2 (please, see Table 3 there).
>
> Please see the additional figure provided in:
> https://docs.google.com/document/d/e/2PACX-1vTI3F6-L4XLqFv9NWiE9LriBkZ-CpIAhvtd9XGDyQr59PAQBv-IG_z8-EaqIQlBXlZwxqeCJNgtqjxA/pub
>
> .
>
> * Answer Q5
>
> Answer in reviewer sTUc Q3 (no space left here)
>
> .
>
> * Answer Q6
>
> We use 5 seeds for training in our experiments, new version is updated with std. All the results of IO-LVM on the test set are statistically better than the respective VAE. Also, we included two more tables to highlight better the outperformance of IO-LVM. Instead of reporting only the reconstruction on the % of match in edges, we are now also reporting the % of match of the full output, i.e., checking if the model output reconstructs perfectly the input, which is a much more difficult task. Please see Tables 1 and 2 in our rebuttal to reviewer Xit2 since there is no space left here.

---

> > ### Comment · Reviewer_2TrQ · 2025-04-04
> >
> > Q1) Thank you for answering this question, I acknowledge that this was not aware to me.
> >
> > Q2) I still do not understand, from your rebuttal, why is x_eps treated to be independent of y or theta, when infact
> > $x_{\epsilon} = argmin E_{\epsilon} [ <y_{\theta} + \epsilon, x> ]$. Does some mathematical trick allow you to do this?
> >
> >
> > Q3) Thank you for answering this question, I acknowledge that this was not aware to me.
> >
> > Q4) While I agree with you that IO-VLM seems like it can make better use of the data using a low dimensional latent space, VAEs might be able to learn better given a higher dimensional latent space. I do not see any result with dimensions greater than 10 in the rebuttal to Xit2. More generally, I acknowledge that I do not know how informative it is to show data efficiency of these seemingly toy problem.
> >
> > Q6) Thank you for reporting this.
> >
> > I will increase my score to weak accept given the rebuttal to my reviews and other reviews combined with the fact that I am not well versed with this general area of research.

---

> > > ### Author Response · Authors · 2025-04-04
> > >
> > > First,we would like to thatnk the reviewer to engage with the rebuttal.
> > > The reviewer has two further remain requests of clarifications that we address below:
> > > * Q2. Why x_eps is intependent of y when taking the gradients.
> > > * Q4. VAEs could potentially be better with more dimensions.
> > >
> > >
> > > Yes, it is a mathematical trick. While a formal mathematical treatment can be found in [Learning with Fenchel-Young Losses, 2019] and [Learning with Differentiable Perturbed Optimizers, 2020], here we provide an intuitive explanation to illustrate why this is true with an example:
> > >
> > > .
> > >
> > > * Answer for Q2.
> > >
> > > **What do we want to show?**
> > >
> > > We aim to show that the partial derivative of the following expression:
> > >
> > > $\langle y^{\theta} + \epsilon, \hat{x}_{\epsilon}^{\theta} \rangle$
> > >
> > > with respect to $y^{\theta}$  is simply $\hat{x}_{\epsilon}^{\theta}$.
> > >
> > > The derivative of $y^{\theta}$  with respect to $\theta$  itself is not relevant here, as this derivative is accounted in a separate term by the neural network's gradient as shown in Eq.6 in the paper (i.e., $\partial{g_{\theta}}$..). Here, we focus exclusively on why the derivative with respect to the argument $y^{\theta}$ of the inner product is exactly  $\hat{x}_{\epsilon}^{\theta}$.
> > >
> > > **Simplifying the notation**
> > >
> > > By using change of variable $y' = y^{\theta} + \epsilon$, our original problem simplifies to showing that $\frac{\partial}{\partial y'} \langle y', \hat{x} \rangle$ is exactly $\hat{x}$ and doesn't depend on $y'$, where as you already mentioned, $\hat{x} = \arg\min_{x \in \mathcal{X}} \langle y', x \rangle$.
> > >
> > > **An example for intuition**
> > >
> > > Let us fix $y' = [8, 3, 5, 9, 15]$, and that we have a linear funtion defined by $\langle y', x \rangle$. In a minimization problem to pick the smallest value in a feasible set (e.g., pick the path with lowest cost), the result using our example is $\hat{x} = [0, 1, 0, 0, 0]$, because the second element is the smallest. And this leads to the minimum value of the function to be $\langle y', \hat{x} \rangle = 3$
> > >
> > > **The partial derivative**
> > >
> > > Now we can evaluate $\frac{\partial}{\partial y'} \langle y', \hat{x} \rangle$ is exactly $\hat{x}$ by slightly changing the elements of $y'$.
> > >
> > > If we slightly perturb the first element (8 → 8.001), the minimum value of $\langle y', \hat{x} \rangle$ remains 3, because it still chooses the second element.
> > >
> > > If we slightly perturb the second element (3 → 3.001), the minimum immediately changes from 3 to 3.001.
> > >
> > > Perturbing other elements (5, 9, 15) does not affect the minimum since they are not chosen.
> > >
> > > **What does it mean?**
> > >
> > > The value of $\langle y', \hat{x} \rangle$ changes only if we perturb the coordinate of $y'$ associated with the optimal solution (the chosen element). This change is linear due to the linearity of the inner product. Therefore, the derivative vector (the gradient) is exactly [0,1,0,0,0], which is equal to $\hat{x}$
> > >
> > >
> > > .
> > >
> > > * Answer for Q4.
> > >
> > > We believe this question is essential to address and add value to the paper. We go further and try to answer the following: How many latent dimensions, and how many samples in the data the VAE needs to be able to capture the problem feasibility? We decided to quickly run an additional experiment only for the VAE with latent dimension = 100 and considering two different training data sizes: 1000 samples and 10000 samples. Below you find the table results together with the results of Table 3 in the rebuttal for reviewer Xit2.
> > >
> > > | Methods    | Latent dims | Train size = 1000 | Train size = 10000 |
> > > | -------- | ------- | ------- | ------- |
> > > | VAE  | 10    | 0.8 \%  | 11.8 \% |
> > > | VAE | 100    | 1.3 \% | 30.2 \% |
> > > | IO-LVM  | 10    | 20.8 \%   | 58.3 \% |
> > >
> > > We see that it might happen that, with a lot of training data, the VAE might be able to "memorize" the reconstruction. However, limited amount of training data keeps the VAE results extremely low in terms of reconstruction. This is because the VAE is not reconstructing the solver input, but the solver output, which is a much harder task with a limited amount of samples.
> > >
> > > Thank you again for the additional questions.

---

### Decision · Program_Chairs · 2025-05-01

**Decision:**

Reject

**Comment:**

The paper proposes a representation learning based method for generating feasible solutions for constrained optimization problems.

Reviewers generally appreciated the motivation of the method, but raised concerns about clarity (e.g., jargon) and comparisons with baselines (e.g., not all get access to the same black box solver as the proposed method; limited scope). During the rebuttal, the authors added another baseline and argued that graph-based problems are ubiquitous, and that prior papers published at top venues have experiments that are no more complex than those in the current submission. During the reviewer discussion, reviewers noted that they didn't see any technical errors with the paper, but reiterated that the limited experiments were a concern.

Overall, the current reviewer scores (3 x weak accept, 1 x weak reject) indicate that the paper does have merit; the use of the Fenchel Young loss in particular may be of interest even to researchers in other domain areas. However, this loss itself appears in multiple of the cited prior papers. I also tend to agree with reviewer Xit2 about the experiments being a bit limited and about the use of a solver (which not all prior methods require). I am thus recommending that the paper be rejected.